# Wheat Yellow Mosaic Virus NIb Interacting with Host Light Induced Protein (LIP) Facilitates Its Infection through Perturbing the Abscisic Acid Pathway in Wheat

**DOI:** 10.3390/biology8040080

**Published:** 2019-10-23

**Authors:** Tianye Zhang, Peng Liu, Kaili Zhong, Fan Zhang, Miaoze Xu, Long He, Peng Jin, Jianping Chen, Jian Yang

**Affiliations:** 1School of Forestry and Biotechnology, Zhejiang Agriculture and Forestry University, Hangzhou 310021, China; ZTye1995@163.com; 2State Key Laboratory for Quality and Safety of Agro-products, Institute of Plant Virology, Ningbo University, Ningbo 315211, China; wood319@126.com (P.L.); zhongkaili@nbu.edu.cn (K.Z.); zf950418@163.com (F.Z.); xumiaoze@yeah.net (M.X.); hnndhelong2@163.com (L.H.); PengJ0310@163.com (P.J.)

**Keywords:** wheat yellow mosaic virus, BIFC, ABA signaling pathway

## Abstract

Positive-sense RNA viruses have a small genome with very limited coding capacity and are highly reliant on host factors to fulfill their infection. However, few host factors have been identified to participate in wheat yellow mosaic virus (WYMV) infection. Here, we demonstrate that wheat (*Triticum aestivum*) light-induced protein (*Ta*LIP) interacts with the WYMV nuclear inclusion b protein (NIb). A bimolecular fluorescence complementation (BIFC) assay displayed that the subcellular distribution patterns of *Ta*LIP were altered by NIb in *Nicotiana benthamiana*. Transcription of *TaLIP* was significantly decreased by WYMV infection and *TaLIP*-silencing wheat plants displayed more susceptibility to WYMV in comparison with the control plants, suggesting that knockdown of *TaLIP* impaired host resistance. Moreover, the transcription level of *TaLIP* was induced by exogenous abscisic acid (ABA) stimuli in wheat, while knockdown of *TaLIP* significantly repressed the expression of ABA-related genes such as wheat abscisic acid insensitive 5 (*TaABI5)*, abscisic acid insensitive 8 (*TaABI8*), pyrabatin resistance 1-Llike (*TaPYL1*), and pyrabatin resistance 3-Llike (*TaPYL3*). Collectively, our results suggest that the interaction of NIb with *Ta*LIP facilitated the virus infection possibly by disturbing the ABA signaling pathway in wheat.

## 1. Introduction

Plant RNA viruses are an important cause of agricultural economic losses [1]. It is particularly important to understand the interaction of the virus with plant host factors in the process of infecting plants. Due to their limited coding capacity, RNA viruses recruit some host proteins via direct or indirect interactions to complete their own reproduction and, hence, only survive in living cells [2,3,4]. More and more evidence supports the idea that versatile host factors are involved in different periods of virus infection. For example, the Potyvirus VPg protein may selectively interact with eukaryotic initiation factor 4E (eIF4E) or its isoform to regulate virion disassembly via potyviral genome translation, Brome mosaic virus (BMV) 1a can interact with reticulon homology proteins (RHPs) to re-localize them from peripheral ER tubules to the interior of the spherules, and glyceraldehyde-3-phosphate dehydrogenase (GAPDH), a crucial host factor, performs an uncanonical function in TBSV replication via interacting with p92 protein [5,6,7]. Therefore, identifying these host factors at the molecular level and understanding their functions in the process of viral infection could help to develop new disease resistance strategies and reveal novel plant antiviral mechanisms [3].

Potyviruses include many agriculturally and economically important pathogens such as Wheat yellow mosaic virus (WYMV), which belongs to the genus *Bymovirus* (*Potyviridae*) [8,9]. The WYMV genome consists of two positive single RNA strands, RNA1 (7.5 kb) and RNA2 (3.6 kb), which each encode a polyprotein [10]. The polyprotein encoded by RNA1 produces eight proteins including the coat protein (CP) and a nuclear inclusion b (NIb) protein that functions as an RNA-dependent RNA polymerase (RdRp), which is important for virus replication. RNA2 encodes a polyprotein (101 kDa) that gives rise to two proteins of 28 kDa (P1) and 73 kDa (P2) [10]. 

The potyviral NIb is a nuclear targeting protein and is a vital part of the virus infection process. For example, the (Turnip mosaic virus) TuMV NIb protein interacts with SCE1 and the interaction is necessary for virus infection [11]. Furthermore, SUMOylation of TuMV NIb promotes virus infection by counteracting the NPR1-mediated resistance pathway [12]. NIb is present in viral replication complexes and is capable of recruiting host proteins such as HSP70, *Nb*SCE1, *At*RH8, and *At*RH9 via interactions to promote potyviral RNA replication [11,13,14,15]. Furthermore, NIb from several other potyviruses can be recognized by dominant resistance genes (*R* genes) such as *Pvr1*, *Pvr2*, *Pvr4*, *Pvr8*, and *Pvr9*, which encode resistance proteins and trigger a Pvr-mediated hypersensitive response [4,16,17,18,19,20]. Nevertheless, WYMV NIb is not only one of the most conserved portions in the WYMV genome, but is also a good candidate for broad spectrum resistance [21]. Indeed, transgenic wheat containing the antisense virus NIb has durable field resistance to WYMV [21]. In addition, WYMV P1 proteins can interact with NIb and recruit it into P2-induced aggregates through its association with P1 [22]. However, we still do not fully understand the function of WYMV-NIb during the process of viral infection.

To investigate the potential roles of NIb during WYMV infection, we used NIb as a bait to screen a wheat (*Triticum aestivum*) yeast cDNA library. We obtained light-induced protein (*Ta*LIP) (accession number: AK454210.1), which is a member of the Fibrillin family in wheat, and verified the interaction between *Ta*LIP and NIb in vivo. A bimolecular fluorescence complementation (BiFC) assay indicated that the interaction between *Ta*LIP and NIb affects their sub-cellular distribution. Furthermore, the transcription level of *TaLIP* was downregulated in WYMV-infected wheat and the *TaLIP* gene silenced wheats were more susceptible to WYMV in comparison to the control wheat plant. Quantitative real-time PCR (qRT-PCR) showed that *TaLIP* is responsive to external ABA stimuli and silencing the *TaLIP* downregulated the transcription level of ABA signaling genes such as *TaNCED*, *TaNCED2*, *TaABI5*, *TaABI8*, *TaPYL1*, *TaPYL3,* and *TaPYL5*. Our results provide a basis for exploring the molecular mechanisms of WYMV pathogenicity in wheat and for providing candidate genes to develop plant transgenic disease resistance.

## 2. Materials and Methods

### 2.1. Plant Materials and Plasmids

*Nicotiana benthamiana* plants were grown in a glasshouse at 23 °C with a 16 h light/8 h dark photoperiod. WYMV-infected seedlings of Yangmai 158 with typical mosaic symptoms were collected from a diseased nursery in Yantai City, Shandong Province, China. 

### 2.2. Phylogenetic Analysis and Promoter Cis-Acting Element Prediction Analysis

Using TaLIP gene and the representative fibrillin genes mainly from *Arabidopsis thaliana, Oryza sativa, Triticum* aestivum, and several other plants to construct the evolutionary tree. Then, we classified them using a previously reported method of *Arabidopsis thaliana* fibrillin proteins [23]. The construction of the evolutionary tree utilizes MEGA 4.0 software [24].

For the promoter prediction analysis, we obtained the sequence of about 2000 bp before the translation initiation site of TaLIP from the wheat genome database (NCBI), then input this sequence into the PlantCARE database [25] for the promoter prediction analysis.

### 2.3. Yeast Two-Hybrid Assay

Yeast two-hybrid assays were performed following the method described in the Takara protocol handbook. The full length of *TaLIP* (accession number: AK454210.1) and WYMV NIb were cloned and fused to the Gal DNA-binding domain (vector: pGBKT7) or Gal4 activation domain (vector: pGADT7), respectively, using primers listed in Appendix A. Yeast cells (strain Y2H Gold) carrying the co-transformed plasmids were plated onto a low-stringency selective medium lacking tryptophan and leucine (SD/-Trp-Leu) to confirm the transformation and plated onto a high-stringency selective medium lacking tryptophan, leucine, histidine, and adenine (SD/-Trp-Leu-His-Ade) to analyze the interaction.

### 2.4. Sub-Cellular Localization, Bimolecular Fluorescence Complementation and Co-Immunoprecipitation (Co-IP) Assays 

For subcellular localization analyses and BiFC, a series of recombinant plasmids including NIb-GFP, *Ta*LIP-GFP, NIb-nYFP, and *Ta*LIP-cYFP were constructed using Gateway technology according to the manufacturer’s instructions (Invitrogen). The first PCR used the primer pairs NIb-GFPN/NIb-GFPC (NIb for localization), *Ta*LIP-GFPN/*Ta*LIP-GFPC (*Ta*LIP for localization), NIb-nYFPN/NIb-nYFPC (NIb for BiFC), and *Ta*LIP-cYFPN/*Ta*LIP-cYFPC (*Ta*LIP for BiFC) (Appendix A). The second PCR was performed using primers attB1 and attB2 (Appendix A) and the amplified product of the first PCR as a template. The amplified product was then introduced into pDONR207 by the BP reaction and the entry clones pENTR-NIb-GFP, pENTR-*Ta*LIP-GFP, pENTR-NIb-nYFP, pENTR-*Ta*LIP-cYFP, and pENTR-*Ta*LIP-HA were constructed. Finally, the LR-clonase reaction was used to transfer NIb and *Ta*LIP fragments from the entry clones to the destination vector and recombinant plasmids including pGWB5C-NIb, pGWB5C-*Ta*LIP, pGTQL1221-NIb, pGTQL1211-*Ta*LIP, and *Ta*LIP-HA were constructed. These recombinants were used to transform the competent *E.coli* strain *DH5α* using heat shock and selected on a medium containing 50 μg/mL kanamycin and 50 μg/mL hygromycin.

The recombinant binary constructs were introduced into *Agrobacterium tumefaciens* strain GV3101 by electroporation (Bio-Rad Gene Pulser, 0.2 cm cuvettes, 25 micro F, >2.1 kV). Agroinfiltration was performed as described by [4]. Briefly, cultures of GV3101 containing a relevant binary plasmid were grown in yeast extract tryptone (YEP) medium containing rifampicin (50 μg/mL) and kanamycin (100 μg/mL) at 28 °C for 16 h.

For sub-cellular localization, *Agrobacterium* cultures containing pGWB5C-NIB and pGWB5C-*Ta*LIP were centrifuged for 30 s at 8000 rpm, resuspended, and then diluted to an OD_600_ of 0.6 (10 mM MES, pH 5.6, 10 mM MgCl_2_, 200 mM acetosyringone) before leaf infiltration. The expression of fluorescent proteins was examined at 72 h post agroinfiltration.

For the BiFC assay, GV3101 strains containing the BiFC plasmids with pGTQL1221-NIb-nYFP and pGTQL1211-*Ta*LIP-cYFP were resuspended and adjusted to an OD_600_ in a 1:1 ratio with infiltration medium before leaf infiltration. The combinations pGTQL1221-NIb-nYFP/pGTQL1211-GUS-cYFP and pGTQL1221-GUS-nYFP/pGTQL1211-*Ta*LIP-cYFP were used as negative controls. The cell suspensions were incubated at room temperature for 2 h to 4 h and then used to infiltrate 5- to 6-week-old *N. benthamiana* leaves. The expression of fluorescent proteins was examined at 72 h post agroinfiltration [26].

For the in vivo co-IP analysis, about 0.5 g Agro-infiltrated leaf tissue frozen in liquid nitrogen was ground to a fine powder and thawed in plant protein extraction buffer containing 10% glycerol, 25 mM Tris-HCl, pH 7.5, 1 mM EDTA, 150 mM NaCl, 2% polyvinylpolypyrrolidone (PVPP), 10 mM DTT, 1× protease inhibitor cocktail (Sigma, Shanghai, China), 0.2% Triton X-100 (Sigma-Aldrich, St. Louis, MO, USA) (1 g tissue per sample/2 mL buffer). The mixture was centrifuged at 18,000 g for 10 min at 4 °C. Each supernatant (500 µL) was mixed with 45 µL anti-GFP conjugated agarose beads (Sigma) and incubated at 4 °C for 1.5 h with gentle shaking. Agarose beads were pelleted and washed three times with the co-IP buffer (10% glycerol, 25 mM Tris-HCl, pH 7.5, 1 mM EDTA, 150 mM NaCl, 2% PVPP, 1 mM DTT, 0.1% Triton X-100). The resulting pellets were mixed individually with SDS loading buffer boiled at 100 °C for 8 min. For immunoblot, proteins were separated in 10% SDS-PAGE gels through electrophoresis, and then transferred to NC membranes. The blots were probed with an anti-HA (1:5000), anti-GFP (1:5000), followed by an HRP-conjugated secondary antibody. The detection signals were developed using an electrochemiluminescence (ECL) reagent as instructed (Thermo Scientific, Hudson, NH, USA), and visualized using a Bio-Rad ChemiDoc Touch imaging system (Bio-Rad, Hercules, CA, USA).

### 2.5. Plant RNA Isolation and Quantitative Real Time PCR Analysis

Leaves were collected from infected wheat plants, frozen, and stored at −80 °C until use. Total RNAs were extracted from plants using Trizol reagent (Invitrogen) and stored at −80 °C. Quantitative real time (qRT)-PCR analysis was performed using an ABI7900HT Sequence Detection System (Applied Biosystems, CA, USA) with an AceQ qPCR SYBR Green Master Mix (Vazyme, Nanjing, Jiangsu, China). At least three biological replicates, with three technical replicates, were used for each assay. The *Triticum aestivum cell division cycle (CDC)* gene (Accession Number: XM_020313450) was used as the internal reference gene for analysis to calculate the fold changes in gene expression. The fold changes were calculated using the 2^-ΔΔC(t)^ method [27]. All gene-specific primers for qRT-PCR are shown in Appendix A.

### 2.6. Virus-Induced Gene Silencing

Barley stripe mosaic virus (BSMV)-based gene silencing vectors were kindly provided by Dr Dawei Li, China, and have been widely used in barley and wheat [28]. *Ta*LIP (200 bp) from the leaf cDNA of wheat was amplified by RT-PCR (Appendix A), inserted into the BSMV γ gene (BSMV:*Ta*LIP), and digested with Pac I and Not I restriction enzymes. Additionally, the BSMV:00 was used as the negative control.

### 2.7. Mechanical Friction Inoculation of Barley stripe mosaic virus and *Wheat yellow mosaic virus*

Virus in vitro transcription followed by friction inoculation was performed as previously described [29,30]. Briefly, in vitro transcription of linearized plasmid transcripts of BSMV RNA α, β, and γ in a molar ratio of 1:1:1 were mixed with an equal amount of excess inoculation buffer (named as FES) (0.06 M potassium phosphate, 0.1 M glycine, 1% bentonite, 1% sodium pyrophosphate decahydrate, 1% celite, pH 8.5) and then inoculated into leaves of 7–10-day-old wheat seedlings. In vitro transcription of linearized plasmid transcripts of WYMV RNA R1 and R2 were also mixed at a molar ratio of 1:1, then inoculated into the upper leave of the BSMV-infected wheat seedling.

## 3. Results

### 3.1. TaLIP Interacts with Wheat Yellow Mosaic Virus NIb and C-Terminus of NIb^196–380aa^ Is the Major Region for This Interaction 

NIb plays an important role in the process of virus infection. To investigate the host factors that interact with the NIb, we used NIb as a bait to screen a wheat (*Triticum aestivum*) yeast cDNA library. A series of proteins were identified via the yeast two-hybrid screening technology including two clones of partial fragment of chlorophyll a-b binding protein 50, eight clones of partial fragment of light-induced protein, and five clones of full length of GDP-L-galactose phosphorylase (Appendix A). Consider the number of selected proteins, the light-induced protein was selected for further investigation. The sequence alignment between *Triticum aestivum* light-induced protein and its homologous genes of *Nicotiana tabacum*, *Capsicum baccatum*, *Helianthus annuus*, and *Lactuca sativa* indicated that the homology was 83.23% (Appendix A). For convenience, we named *Triticum aestivum* light-induced protein as *Ta*LIP in this study. Subsequently, yeast two-hybrid assays also demonstrated an interaction between *Ta*LIP and WYMV NIb (Figure 1a). Furthermore, to determine the key domain for interaction, WYMV NIb (380 aa) was further divided into two smaller fragments, NIb^1–195^ (encoding aa 1–195) and NIb^196–380^ (encoding aa 196–380) for yeast two-hybrid assays (Figure 1b). Although both NIb^1–195^ and NIb^196-380^ interacted with *Ta*LIP, the yeast two-hybrid assays revealed a stronger interaction between NIb^196–380^ and *Ta*LIP (Figure 1c). Thus, residues 196 to 380 of NIb appear to be crucial for its interaction with *Ta*LIP.

### 3.2. TaLIP Interacts with WYMV NIb In Vivo

To test the interaction between NIb and TaLIP in vivo, co-immunoprecipitation (Co-IP) assays were conducted. For the Co-IP assays, GFP-tagged NIb (NIb-GFP) was transiently co-expressed with HA-tagged *TaLIP* (*Ta*LIP-HA) and GFP-tagged was transiently co-expressed with *Ta*LIP-HA in *N. benthamiana* leaves as the control. Leaf tissues were then collected at 60 h post infiltration (hpi). Total protein extracts were immunoprecipitated using anti-GFP antibody coupled to agarose beads, and the resulting precipitates were analyzed by immunoblot using anti-HA antibodies. We observed that *Ta*LIP co-immunoprecipitated with NIb-GFP, but not with GFP alone (Figure 2). Taken together, these results further demonstrate an interaction between WYMV NIb and *Ta*LIP in vivo.

### 3.3. Sub-Cellular Localization of WYMV NIb and TaLIP Is Altered by Their Interaction in Nicotiana Benthamiana

To determine whether the sub-cellular localization of *Ta*LIP and NIb was affected by the interaction of these proteins, the recombinant plasmids expressing *Ta*LIP or WYMV NIb fused with Green fluorescent protein (GFP) at their C terminus (*Ta*LIP:GFP and NIb:GFP) were constructed and introduced into *N. benthamiana* epidermal cells by *Agrobacterium* infiltration. At 72 h post infiltration (hpi), GFP fluorescence was detected by confocal microscopy. *Ta*LIP:GFP was observed in the chloroplast and NIb:GFP was observed in the nucleus and cytoplasm (Figure 3). Subsequently, we performed BiFC assays to analyze the interactions between *Ta*LIP and WYMV NIb. pGTQL1211-GUS-cYFP and pGTQL1221-GUS-nYFP were used as the two negative controls, respectively. These combinations of fusion proteins were all expressed in *N. benthamiana* via agroinfiltration. When pGTQL1221-NIb-nYFP (YN-NIb) and pGTQL1211-*Ta*LIP-cYFP (YC-*Ta*LIP) were co-expressed in *N. benthamiana* epidermal cells, some aggregates were observed in the cytoplasm and strong GFP signals (we changed the pseudo-color of the YFP signal to green for convenience) were observed around the chloroplast, but not in the chloroplast. In contrast, no significant fluorescence signal was detected in the negative controls (Figure 4). It is interesting that the sub-cellular localization of *Ta*LIP and WYMV NIb were completely different from that observed when performing the BiFC assay. Based on these data, we speculate that the interaction between NIb and *Ta*LIP may affect the distribution patterns of WYMV NIb and *Ta*LIP.

### 3.4. The Transcriptional Level of TaLIP Is Downregulated in WYMV-Infected Wheat

To clarify whether *TaLIP* was affected at the transcriptional level by WYMV infection, we then designed primer pairs for quantitative RT-PCR (qRT-PCR) and characterized the expression profile of *TaLIP* in *Triticum aestivum* with or without WYMV infection. The expression level of *TaLIP* was significantly downregulated 0.23-fold in WYMV-infected wheat when compared to that of WYMV-uninfected wheat (Figure 5). These results indicate that the expression of *TaLIP* in the WYMV-infected wheat plants can be regulated, potentially to benefit WYMV infection in this host.

### 3.5. Knockdown the TaLIP Facilitate WYMV Infection in Wheat

To investigate the relationship between *TaLIP* expression and WYMV infection in wheat, we inoculated six wheat seedling with an RNA transcript representing BSMV + WYMV or BSMV:TaLIP + WYMV. After 7 dpi, we analyzed the silencing level of the *TaLIP* gene in the BSMV:TaLIP + WYMV co-inoculated wheat seedling through qRT-PCR using *TaLIP* specific primers. The results showed that the *TaLIP* transcript level in the plants co-inoculated with BSMV:TaLIP + WYMV were better silenced (*p* < 0.01) than the plants co-inoculated with BSMV + WYMV (Figure 6a). Then, the expression level of WYMV CP was also detected by qRT-PCR using the CP specific primers in these plants. The results showed that the expression level of WYMV CP was detected by qRT-PCR and the WYMV CP expression level of BSMV:*Ta*LIP + WYMV inoculated wheat was significantly higher than the inoculated wheat (BSMV + WYMV) (Figure 6b). These results suggest that knockdown of *TaLIP* impaired host resistance.

### 3.6. TaLIP Belongs to the FBN1 Subspecies of the Fibrillin Family and Has Three Copies of ABA Responsive Promoter Element

Based on the above findings, we can speculate that NIb interacting with TaLIP might facilitate the WYMV infection by perturbing the pathway mediated by *TaLIP* in wheat. To predict the potential function of TaLIP in the process of WYMV infection, the phylogenetic relationship of *Ta*LIP between other FBN proteins was constructed. Fourteen *Arabidopsis thaliana* FBN proteins, eight *Oryza sativa* FBN proteins, 12 *Triticum aestivum* FBN proteins, and nine other plant FBN proteins were selected for further analyses. The phylogenetic analysis was performed using the MEGA4 program [24]. Phylogenetic analysis indicated that *Ta*LIP was categorized into subfamily FBN1 (Figure 7a). There is some evidence that the plant hormones regulate FBN gene expression in plants such as ABA [31]. Interestingly, the Cis-acting element information of the *TaLIP* gene showed that there were nine copies of the ABA responsive promoter element (Figure 7b). We predicted that the *TaLIP* gene is involved in the response to ABA.

### 3.7. TaLIP Is Responsive to External ABA Stimuli, Silencing the TaLIP Gene Suppressing the ABA Signaling Pathway

To determine how *TaLIP* responds to exogenous applications of hormones, we investigated the transcription profiles of *TaLIP* in wheat treated with ABA. After ABA treatment, expression was induced from 0.5 to 12 h post-treatment (hpt), rapidly increased to 2.15-fold at 0.5 hpt, reached a peak at 6 hpt (about 3.29-fold that of nontreatment), then decreased to 2.60-fold higher than the nontreated control (N) from 6 to 12 hpt (Figure 8a). These results suggest that *TaLIP* transcription was induced by exogenous ABA stimuli. Next, we investigated the expression of mRNA level of the ABA-related genes including the ABA biosynthetic pathway gene *TaNCED, TaNCED2,* and the ABA signaling pathway gene *TaABI5, TaABI8, TaPYL1, TaPYL3,* and *TaPYL5* in *TaLIP*-silenced wheat (Figure 8b). The transcription level of *TaABI5*, *TaABI8*, *TaPYL1,* and *TaPYL3* were suppressed in *TaLIP*-silenced wheat.

## 4. Discussion

Numerous studies have demonstrated that to cause an infection in plants, virus have evolved to encode factor(s) to defeat plant defense machinery. As stated in the introduction, the potyviral NIb is a vital protein response for virus infection. However, we still do not fully understand the function of WYMV-NIb during the viral infection. In this study, we obtained a fibrillin (FBN) protein, *Ta*LIP, by performing yeast two-hybrid screening, that could interact with NIb and the C-terminal of NIb was the key domain for this interaction (Figure 1). Then, the co-IP assay confirmed the interaction between NIb and *Ta*LIP (Figure 2). FBNs are a large protein family present in photosynthetic organisms ranging from cyanobacteria to higher plants [32], and FBNs are involved in plant responses to biotic stress [23]. Previous study has demonstrated that the NIb of several potyvirals can interact with host protein(s) to promote viral infections such as HSP70, *Nb*SCE1, and *At*RH9 [11,13,15]. Additionally, recent research has shown that fibrillin/CDSP 34 (FBN1) protein levels decreased in tobacco plants infected with tobacco mosaic virus (TMV), which supported our result that the *TaLIP* transcription level of WYMV-infected wheat was significantly downregulated when compared to that of WYMV-uninfected wheat (Figure 5). Therefore, we predicted that *TaLIP* is possibly involved in the process of WYMV infection. Some researchers have proven that FBNs are involved in disease resistance. For example, knockdown of expression of the *LeChrC* (FBN1) gene in tomato caused greater susceptibility to *Botrytis cinereal;* knockdown of FBN4 expression in apple and a mutation of *FBN4* in *Arabidopsis* caused greater susceptibility to the pathogenic bacteria *Erwinia amylovora* and *Pseudomonas syringae pv. tomato*, respectively; and a mutant of the *FBN1b* in *Arabidopsis* was more susceptible to *P. syringae pv. Maculicola* [31,33,34]. Consistent with these findings, knockdown of *TaLIP* expression facilitated WYMV infection in wheat (Figure 6). According to these data, we suggest that Nib–*Ta*LIP interaction possibly promotes WYMV infection. Many FBNs have been previously reported to be chloroplast related proteins such as cucumber CHRC (FBN1) [35]. Consistently, sub-cellular localization analysis showed that *Ta*LIP was also located in chloroplast (Figure 3). Previous study has demonstrated that the NIb of several other potyvirals was located in the nucleus and cytoplasm [12]. Consistent with this, sub-cellular localization analysis of NIb showed that NIb was also located in the nucleus and cytoplasm (Figure 3). Interestingly, BiFC assay revealed a fact that the NIb-*Ta*LIP interaction completely changed their sub-cellular localization in *N. benthamiana* epidermal cells and a large number of fluorescent signals around the chloroplast, but not in the chloroplast (Figure 4). It is possible that Nib–*Ta*LIP interaction affects the function of *Ta*LIP through changing the sub-cellular localization to facilitate WYMV infection. This assumption is supported by findings showing that TuMV NIb interacting with *Nb*EXPA1 recruits *Nb*EXPA1 to the viral replication complex and promotes TuMV infection [9]. Early study has shown that the negatively regulated ABA responses abscisic acid insensitive 2 (ABI2) interacts with the *Arabidopsis* FNB1a to regulate its localization, leading to perturbing the ABA pathway [36]. In addition, ABA appears to enhance plant antiviral defense as shown for several viruses [37,38,39]. Hereby, we reasonably speculate the interaction between the NIb and TaLIP facilitated the WYMV infection by affected the ABA pathway associated with TaLIP. Indeed, there is some evidence that hormones can regulate FBN gene expression [23]. For example, in chromoplasts of bell pepper fruit, indole-3-acetic acid can delay the accumulation of FBN protein, whereas ABA can accelerate it [40]. Indeed, the promoter Cis-acting element assay of *TaLIP* showed that it has three copies of the ABA responsive promoter element (Figure 7b), and the expression of the *TaLIP* gene was significantly induced by external ABA stimuli (Figure 8a). Thus, we speculate that *TaLIP* is a responsible gene involved in the ABA signal pathway. Consistent with this notion, the mRNA expression level of ABA pathway genes *TaABI5*, *TaABI8*, *TaPYL1*, and *TaPYL3* were downregulated in *TaLIP*-silenced wheat (Figure 8b). *TaABI5* and *TaABI8* were reported as positive modulators of ABA signaling and *PYL 1* and *PYL3* were reported to be ABA receptors positively responsive to ABA [41]. While transcriptome analysis revealed that the transcription levels of *TaABI5*, *TaABI8*, *TaPYL1*, and *TaPYL3* were also significant lower in WYMV-infected wheat when compared to that of healthy wheat (Appendix A). In the last few decades, ABA has been reported as a key hormone involved in tuning responses to several abiotic stresses and also induces different resistance mechanisms to viruses, regardless of the induction time [33]. Taken together, our work revealed that WYMV NIb interacts with host *Ta*LIP to promote the WYMV infection possibly through affecting the ABA signal pathway.

## 5. Conclusions

In this study, we analyzed the function of the *Ta*LIP gene in the process of WYMV infection. The sub-cellular localization changes of *Ta*LIP and NIb under the interaction indicated that the *Ta*LIP–NIb interaction may affect the function of *Ta*LIP. Furthermore, the result of the *TaLIP* knockdown experiment indicated that *Ta*LIP may act as a host protein interacting with WYMV NIb to facilitate WYMV infection. External ABA stimuli and the ABA pathway gene expression of *Ta*LIP silenced wheat results indicated that *TaLIP* may be a responsible gene involved in the ABA signal pathway. In summary, our study revealed that WYMV NIb interacts with host *Ta*LIP to promote the WYMV infection possibly through affecting the ABA signal pathway. Additionally, this work provides a basis for exploring the molecular mechanisms of WYMV pathogenicity in wheat and in providing candidate genes to develop plant transgenic disease resistance.

## Figures and Tables

**Figure 1 biology-08-00080-f001:**
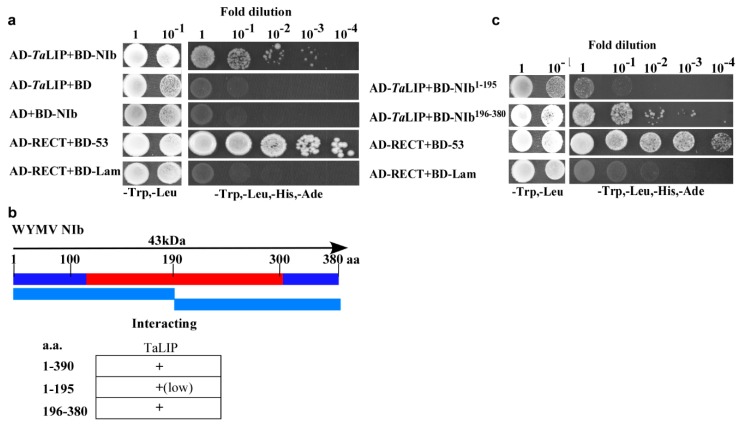
Interaction of Wheat yellow mosaic virus (WYMV)NIb with and *Ta*LIP in a yeast two-hybrid analysis. (**a**) NIb was fused to the DNA-binding domain and *Ta*LIP was fused to the activation domain (AD-*Ta*LIP), co-transformed into yeast cells, and then coated uniformly on selection plates of SD/-Trp, -Leu, -Ade, -His solid medium. Positive and negative controls were co-transformed with AD-T/BK-53 and AD-T/BK-Lam, respectively. (**b**) NIb was divided into two parts and fused to the DNA-binding domain, co-transformed with AD-*Ta*LIP into yeast cells, and then coated uniformly on selection plates of SD/-Trp, -Leu, -Ade, and -His solid medium. AD-T/BK-53 and AD-T/BK-lam were used as the positive control and negative control, respectively. (**c**) Illustration of the WYMV NIb sequence, which divided into two segments and summary of *Ta*LIP interactions with WYMV NIb. The red region represents the conserved domains of NIb.

**Figure 2 biology-08-00080-f002:**
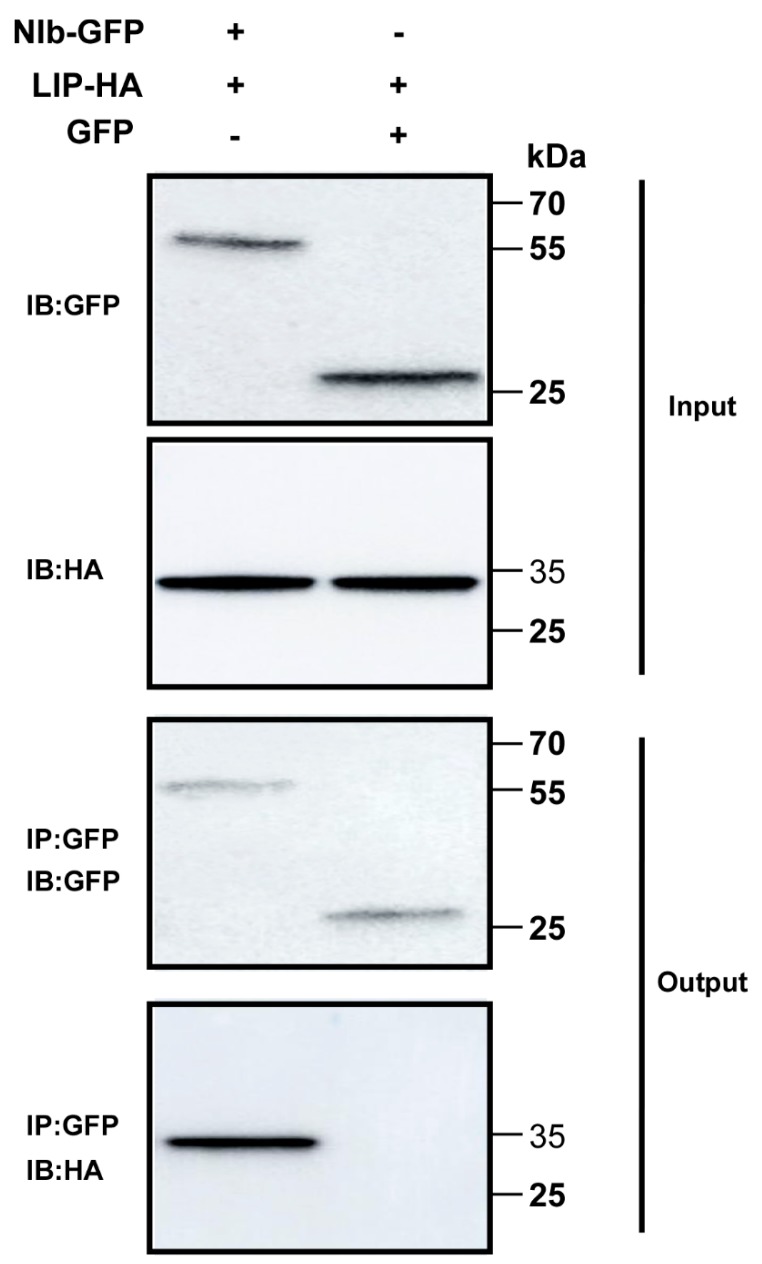
Co-immunoprecipitation analysis of the interaction between NIb and *Ta*LIP. The Green fluorescent protein (GFP) was fused at the C-terminus of NIb (NIb-GFP) and transiently co-expressed with the *Ta*LIP-HA in *N. benthamiana* leaves. GFP-tagged was transiently co-expressed with *Ta*LIP-HA in *N. benthamiana* leaves as the control. The *Ta*LIP-His tag (HA) was used to co-immunoprecipitate with NIb-GFP. The blots were probed with a GFP specific antibody or a HA specific antibody. IP, immunoprecipitation with special antibody. IB, immunoblot with special antibody. The sizes of the proteins in kDa are shown to the left.

**Figure 3 biology-08-00080-f003:**
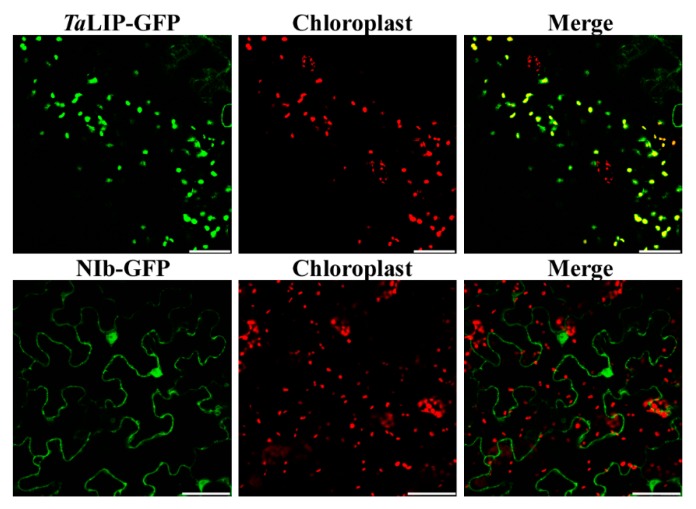
Localization of the *Ta*LIP and WYMV NIb protein in *Nicotiana benthamiana* leaves agroinfiltrated with pENTR-NIb-GFP or pENTR-*Ta*LIP-GFP. The green fluorescent signal of *Ta*LIP proteins was co-localized with the red auto-fluorescent signal of chloroplasts. Fluorescence photographs were taken at 3 dpi. Scale bar: 50 μm.

**Figure 4 biology-08-00080-f004:**
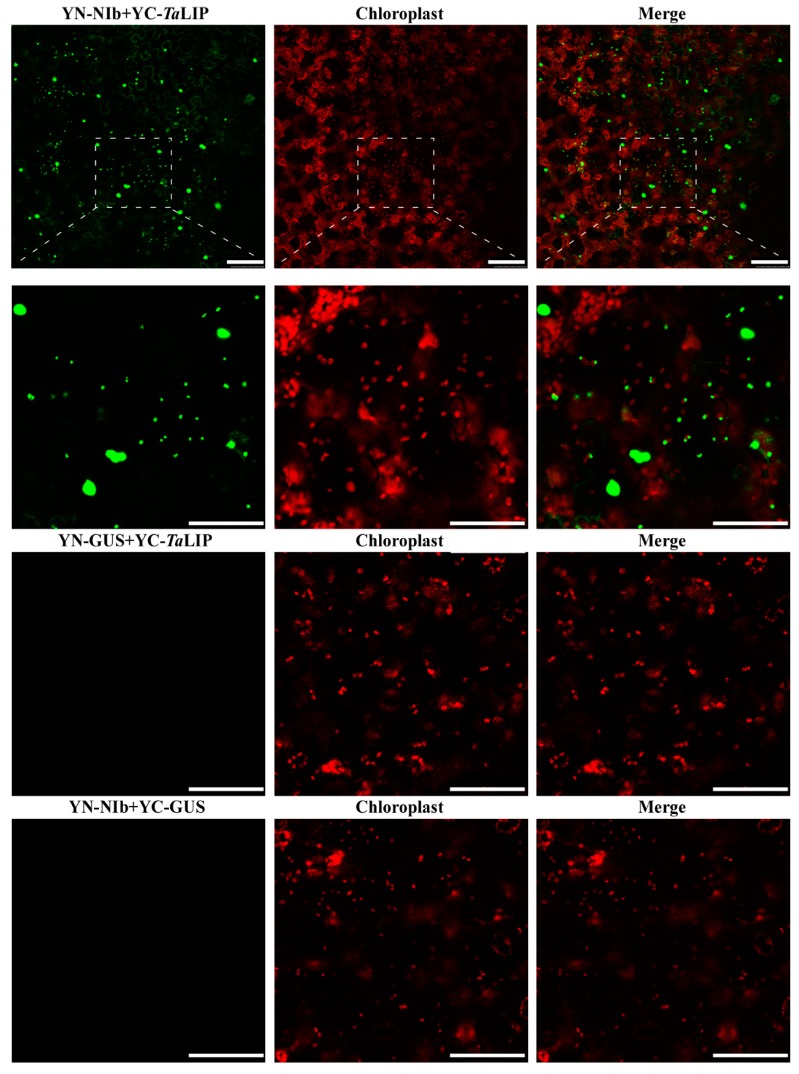
Bimolecular fluorescence complementation (BiFC) assay of the WYMV NIb and *Ta*LIP interactions in *Nicotiana benthamiana*. The pseudo-color of the Yellow fluorescent protein (YFP) signal was changed into green for convenience. The red fluorescent signal is a chloroplast auto-fluorescence signal. β-Galactosidase (GUS)-fused vectors pGTQL1211-GUS-cYFP/pGTQL1221-NIb-nYFP (YN-NIb/YC-GUS) and pGTQL1221-GUS-nYFP/pGTQL1211-*Ta*LIP-cYFP (YN-GUS/YC-*Ta*LIP) are the negative controls. Fluorescence photographs were taken at 3 dpi. Scale bar: 50 μm.

**Figure 5 biology-08-00080-f005:**
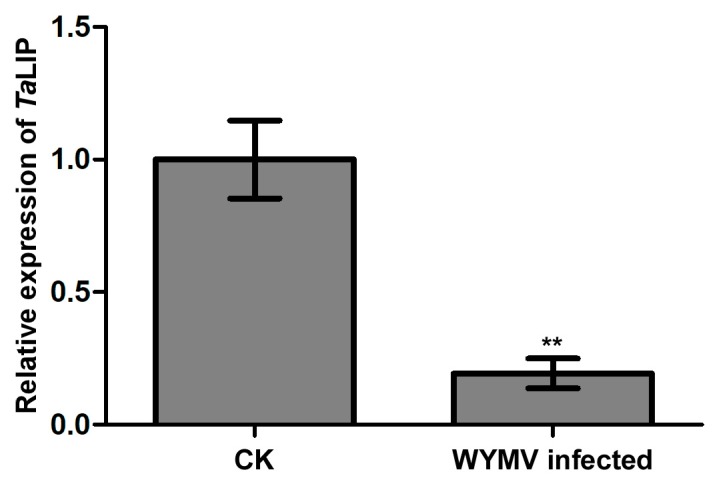
The expression level of *TaLIP* was significantly downregulated in WYMV-infected wheat. *TaCDC* was used as the internal control. The level of *TaLIP* in health wheat was normalized to 1. Each relative expression level is presented as the mean ± SD from four biological samples and each biological sample had four technical replicates. Statistical analyses were done using the Student’s *t*-test. Asterisks indicate a significant difference when compared to the control. * *p* < 0.05; ** *p* < 0.01.

**Figure 6 biology-08-00080-f006:**
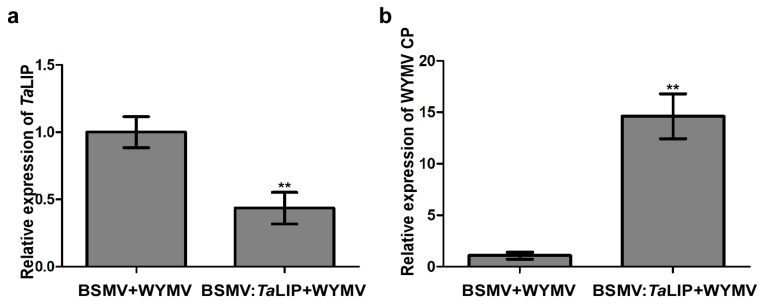
Silencing *TaLIP* gene expression through virus-induced gene silencing (VIGS) significantly promoted WYMV infection in *Triticum aestivum*. (**a**) The quantitative RT-PCR analysis of WYMV CP expression in the inoculated leaves harvested from the *TaLIP*-silenced or non-silenced plants. (**b**) Quantitative RT-PCR analysis of *TaLIP* expression silenced by BSMV-mediated VIGS in *Triticum aestivum*. The *TaCDC* was used as the internal control. Bar represents the SD of three experiments (each with three technical replicates), Asterisks indicate ** *p* < 0.01 to the amount of relative WYMV CP and *TaLIP* expression by the Student’s *t*-test.

**Figure 7 biology-08-00080-f007:**
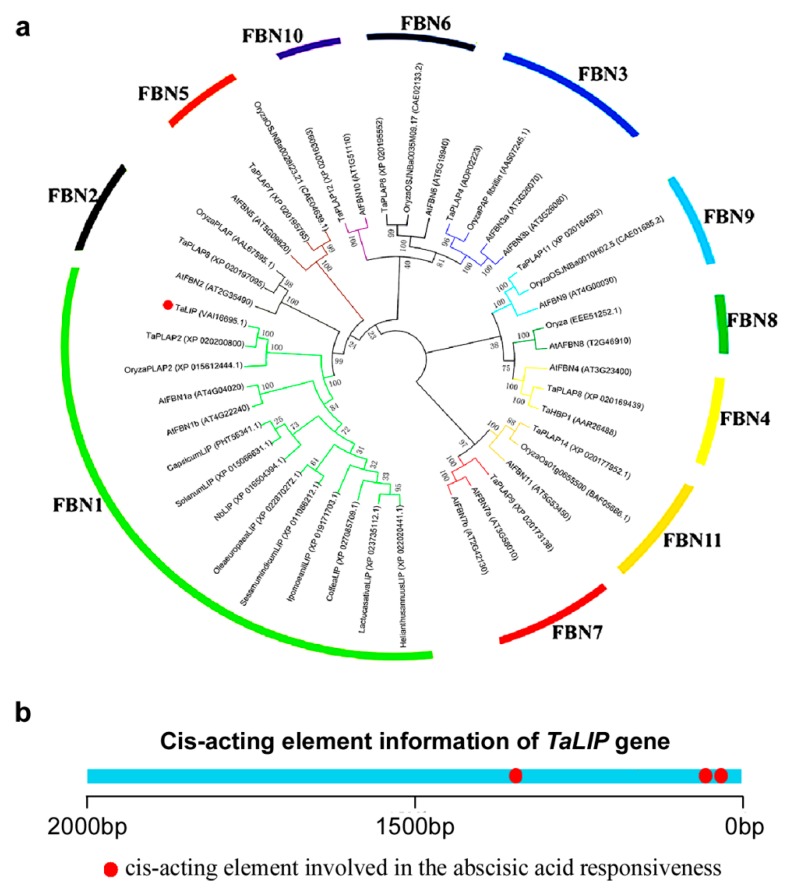
The *Ta*LIP protein belongs to the FBN1 (fibrillin family) and have three copies of the abscisic acid (ABA) responsive element. (**a**) Phylogenetic analysis of *TaLIP* and the construction of the evolutionary tree utilizing MEGA4 program [24]. The 11 distinct subfamilies were designated as 1~11 and labeled with different colored branches respectively. (**b**) promoter Cis-acting element prediction analysis of *TaLIP*.

**Figure 8 biology-08-00080-f008:**
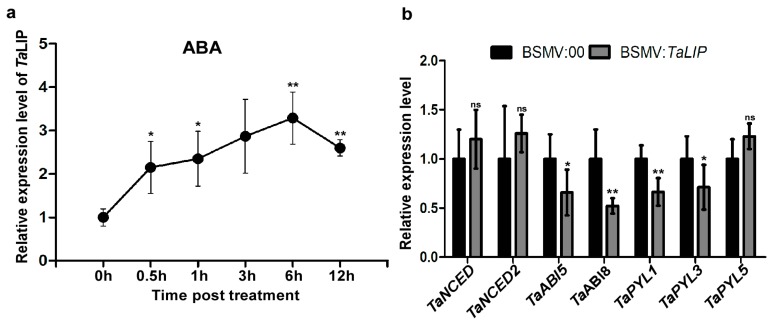
Exogenous ABA treatment induced the expression of *TaLIP* and the ABA signaling pathway was suppressed in *TaLIP*-silenced wheat. (**a**) Expression patterns of *Ta*LIP in wheat after treatment with exogenous hormones ABA for 0.5, 1, 3, 6, and 12 h. Relative expression of *TaLIP* is shown as fold change in transcription over the non-treatment control (0 h). Each relative expression level is presented as the mean ± SD from three biological samples and each biological sample had four technical replicates. Statistical analyses were done using the Student’s *t*-test. Asterisks indicate a significant difference when compared to the control. * *p* < 0.05; ** *p* < 0.01. (**b**) Quantitative RT-PCR analysis of *TaNCED, TaNCED2, TaABI5, TaABI8, TaPYL1, TaPYL3,* and *TaPYL5* expression in the inoculated leaves harvested from the *TaLIP*-silenced or non-silenced wheat. The *TaCDC* was used as the internal control. Each relative expression level is presented as the mean ± SD from three biological samples and each biological sample had four technical replicates. Statistical analyses were done using the Student’s *t*-test. Asterisks indicate a significant difference when compared to the control. * *p* < 0.05; ** *p* < 0.01.

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
