# Peer review of "Wheat Yellow Mosaic Virus NIb Interacting with Host Light Induced Protein (LIP) Facilitates Its Infection through Perturbing the Abscisic Acid Pathway in Wheat"

_biology, 2019, doi:10.3390/biology8040080_

Round 1
Reviewer 1 Report
Zhang et al. analyze the interaction between a host protein (TaLIP) and the NIb of WYMV and the role of this interaction in the outcome of the infection. I think that characterizing host proteins involved in the development of virus infections is always of interest as, despite it is generally accepted the host proteins participate in the development of virus infections, it is an underexplored subject. The paper is generally well written and the results are clearly presented. I only have some minor comments:
1 – My major comment about the paper is the lack of an statistical analyses section in the material and methods. Certainly the figure legends specify the statistical tests used. However, there is no explanation on how these test were selected. For instance, T-tests are only acceptable if the analyzed variables are parametric. Did the authors perform this analysis? The authors should also state clearly the number of replicates in each analysis.
I am unsure of which is the information that the phylogenetic tree provides. As the manuscript is organized, it only interrupts the line of presentation of the molecular analyses. I would suggest deleting it or moving it to a different part of the manuscript unless the authors can justify why they place the tree where it is now. In the same sense. Limiting the explanation of the tree construction to the software used is quite imprecise. For instance, what method was used to build the tree? Which nucleotide substitution model was used, and how was it selected?
Subsections of the Results seem to be organized according to experiment rather than according to conceptual units, which makes the paper more difficult to follow for the reader. I would suggest grouping results according to the evidence they provide. For instance, grouping all the experiments showing the subcellular localization of the proteins together, and putting all the experiments regarding gene expression in another subsection.
I missed additional discussion on the subcellular localization of TaLIP and Nib. Why do the authors think this localization changes? What would be the benefit of this change for the virus or for the host?
L.44. Potyviruses is a common name. As such, it should not be in italics.
The first word in all figure legends should have th first letter in uppercase.
Asterisks in figures are placed randomly. Some times they are in the column with the lowest value, some times in the column with the highest, some times in the right column, some times in the left column… I would suggest homogenizing this.
Author Response
Reviewer 1:
My major comment about the paper is the lack of a statistical analyses section in the material and methods. Certainly the figure legends specify the statistical tests used. However, there is no explanation on how these test was selected. For instance, T-tests are only acceptable if the analyzed variables are parametric. Did the authors perform this analysis? The authors should also state clearly the number of replicates in each analysis.
Answer: Thank you very much for your advices. We have modified these mistakes in this manuscript and marked these in yellow.
I am unsure of which is the information that the phylogenetic tree provides. As the manuscript is organized, it only interrupts the line of presentation of the molecular analyses. I would suggest deleting it or moving it to a different part of the manuscript unless the authors can justify why they place the tree where it is now. In the same sense. Limiting the explanation of the tree construction to the software used is quite imprecise. For instance, what method was used to build the tree? Which nucleotide substitution model was used, and how was it selected?
Answer: Thank you very much for your advices. To predict the potential function of TaLIP in the process of virus infection, the phylogenetic tree was constructed in this manuscript. And there are some evidences have shown that the ABA could regulated the expression level of FBN gene in plant. So the Cis-acting element information of TaLIP also was analyzed. In the view of the organized of this manuscript and the line of presentation of the molecular analyses, we have removed the phylogenetic tree to Figure 7 and 2.7 section.
Subsections of the Results seem to be organized according to experiment rather than according to conceptual units, which makes the paper more difficult to follow for the reader. I would suggest grouping results according to the evidence they provide. For instance, grouping all the experiments showing the subcellular localization of the proteins together, and putting all the experiments regarding gene expression in another subsection.
Answer: Thank you very much for your advices. In order to describe the Results more smoothly, we have modified it according to your suggestion. The modified part has been marked in yellow in the part of Results.
I missed additional discussion on the subcellular localization of TaLIP and Nib. Why do the authors think this localization changes? What would be the benefit of this change for the virus or for the host?
Answer: Thank you very much for your advices. The sub-cellular localization analysis of NIb and TaLIP have shown that the NIb was located in the nucleus and cytoplasm and TaLIP was located in chloroplast (Figure 3). But the BIFC assay revealed that NIb-TaLIP interaction completely changed their sub-cellular localization in Nb epidermal cells, large number of fluorescent signals around the chloroplast but not in the chloroplast (Figure 4) (line 279-285). The negatively regulates ABA responses Abscisic Acid Insensitive 2 (ABI2) interacts with the Arabidopsis FNB1a to regulates its localization lead to perturb the ABA pathway. In addition, ABA appears to enhance plant antiviral defense as shown for several viruses Hereby, we reasonable speculation the interaction between the NIb and TaLIP facilitated the WYMV infection by affected the ABA pathway associated with TaLIP (Line 285-194).
44. Potyviruses is a common name. As such, it should not be in italics.
Answer: Thank you very much for your careful review. We have modified this mistake in this paper according with your suggestion.
The first word in all figure legends should have th first letter in uppercase.
Answer: I so sorry for this mistake. We have modified these mistake according with your advices.
Asterisks in figures are placed randomly. Some times they are in the column with the lowest value, some times in the column with the highest, some times in the right column, some times in the left column… I would suggest homogenizing this.
Answer: Thank you very much for your suggestion. We have revised as your advices in this manuscript.
Thank you very much for your advices again.we have modified these mistakes in this manuscript and marked these in yellow.
In addition Revised supplemental table S1 have been uploaded again.

Reviewer 2 Report
In this manuscript Zhang et al analyze the interaction between wheat yellow mosaic virus NUCLEAR INCLUSION B PROTEIN (NIb) and the wheat host protein LIGHT-INDUCED PROTEIN (TaLIP). The manuscript is well presented, relatively well written and the science presented is consistent. There are a couple of issues with the manuscript that I consider are important to sustain the conclusions. Please find my comments below:
Major comments:
-Can the authors speculate why from their initial screening they select only the TaLIP protein for further investigation? This is not intuitive.
-One of the main issues of the results is that both proteins have a different cellular localization what makes the proposed interaction very complicated to happen in-vivo. TaLIP accumulates in the chloroplast while NIb accumulates in the nucleus. How do the authors explain that NIb can interact with a chloroplastic protein? Is this interaction taking place during nuclear processing of the chloroplastic protein? If so, how is the interaction showing up as cytoplasmic bodies? Furthermore, can the authors explain why TaLIP has two different cellular localizations in Figure 2 (nuclear) and Figure 3 (chloroplastic)?
-Also, FRET is a technique that can be subjected to artifacts. A necessary control for being 100% sure about the interaction between both proteins (shown in Figure 4) would be to use the inverse YN and YC constructs for TaLIP and NIb (YN-TaLIP and YC-NIc).
Minor comments:
Some typos and minor grammar issues in the manuscript like lines 24 and 25: “riesistance” should be “resistance”. There are many more in the manuscript, please double check and correct.
Author Response
Can the authors speculate why from their initial screening they select only the TaLIP protein for further investigation? This is not intuitive.
Answer: Thank you very much for your suggestion. Consider the number of selected protein, the TaLIP protein was selected for further investigation.
One of the main issues of the results is that both proteins have a different cellular localization what makes the proposed interaction very complicated to happen in-vivo. TaLIP accumulates in the chloroplast while NIb accumulates in the nucleus. How do the authors explain that NIb can interact with a chloroplastic protein? Is this interaction taking place during nuclear processing of the chloroplastic protein? If so, how is the interaction showing up as cytoplasmic bodies? Furthermore, can the authors explain why TaLIP has two different cellular localizations in Figure 2 (nuclear) and Figure 3 (chloroplastic)?
Answer: Thank you very much. The sub-cellular localization analysis of NIb and TaLIP have shown that the NIb was located in the nucleus and cytoplasm and TaLIP was located in chloroplast (Figure 3). But the BIFC assay revealed that NIb-TaLIP interaction completely changed their sub-cellular localization in Nb epidermal cells, large number of fluorescent signals around the chloroplast but not in the chloroplast (Figure 4). TaLIP belongs to the FBN1 subspecies of the Fibrillin family. Several studies have characterized the several members of FBN1 was present in chloroplasts with the mature FBN1 protein lacking its chloroplast transit peptide. So we speculate the interaction taking place during nuclear processing of the TaLIP protein. Considering the high artifacts of FRET technique, CO-IP technique was used to replace the experiment in this manuscript. Taken together, the YTH, CO-IP and BIFC experiments demonstrated NIb-TaLIP interaction in vivo.
Also, FRET is a technique that can be subjected to artifacts. A necessary control for being 100% sure about the interaction between both proteins (shown in Figure 4) would be to use the inverse YN and YC constructs for TaLIP and NIb (YN-TaLIP and YC-NIc).
Answer: Thank you very much. Considering the high artifacts of FRET technique, CO-IP technique was used to replace the experiment in this manuscript. Taken together, the YTH, CO-IP and BIFC experiments demonstrated NIb-TaLIP interaction in vivo.
Some typos and minor grammar issues in the manuscript like lines 24 and 25: “riesistance” should be “resistance”. There are many more in the manuscript, please double check and correct.
Answer: Thank you very much. We have modified these mistakes and double check the whole manuscript.
Thank you very much for your advices again. we have modified these mistakes in this manuscript and marked these in yellow.
In addition, revised supplemental table S1 have been uploaded again. Please see the attachment.

Round 2
Reviewer 2 Report
The authors have now addressed all my comments.